# Classification Done Right for Vision-Language Pre-Training

**Zilong Huang    Qinghao Ye    Bingyi Kang    Jiashi Feng    Haoqi Fan**

ByteDance Research

Code & Models: `x-cls/superclass`

## Abstract

We introduce SuperClass, a super simple classification method for vision-language pre-training on image-text data. Unlike its contrastive counterpart CLIP [57] who contrast with a text encoder, SuperClass directly utilizes tokenized raw text as supervised classification labels, without the need for additional text filtering or selection. Due to the absence of the text encoding as contrastive target, SuperClass does not require a text encoder and does not need to maintain a large batch size as CLIP [57] does. SuperClass demonstrated superior performance on various downstream tasks, including classic computer vision benchmarks and vision language downstream tasks. We further explored the scaling behavior of SuperClass on model size, training length, or data size, and reported encouraging results and comparisons to CLIP .

## 1   Introduction

Pretraining methodologies [35, 57, 51, 60] that directly harness web-scale image-text dataset have transformed the field of computer vision in recent years. Among them, contrastive language image pretraining (CLIP) [57] has gained escalading popularity and become predominant due to the following reasons. First, it serves as the industry-standard pre-trained model that facilitates zero-shot visual recognition [50, 52] and finetuning on downstream tasks [19, 17]. Second, proper scaling behaviors [12] are observed such that CLIP can consistently benefit from larger models and bigger data to some extent. Third, it offers strong cross-modal abilities as it is inherently designed to understand and connect information across text and images. Therefore, CLIP-style models are the default choices for most modern Visual Language Models [47, 2, 1], which connect a vision backbone with a deep language model [69, 13].

Despite its success, CLIP necessitates very large batch sizes for training—typically over 64,000—to achieve optimal performance, along with substantial computational resources for text encoding. This high computational demand limits accessibility for researchers with limited resources and engineering expertise. In our work, we aim to address the heavy computational burden by replacing contrastive methodology with a simpler classification approach, eliminates the need for large contrastive batch sizes, and text encoders.

In this work, we revisit the classification method for pretraining on large-scale text-image pairs. Some previous works [54, 31, 39, 27, 51] attempt to tackle this by employing bag-of-words classification in a weak supervised learning manner. However, most of these studies have been conducted on a small scale, and there is no evidence demonstrating their scalability in terms of data and model size. In contrast, our method demonstrates the performance of SuperClass on a scale comparable to CLIP [57], achieving favorable model performance with 13 billion seen samples on 1 billion unique text-image pairs. Some other concurrent efforts [31] have also attempted to replace contrastive learning with classification. However, they rely heavily on preprocessing the text modality, using

38th Conference on Neural Information Processing Systems (NeurIPS 2024).

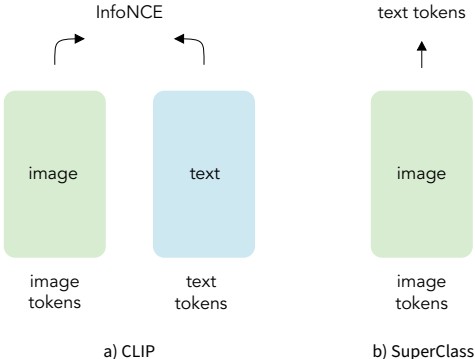

Figure 1: (left) CLIP uses two separate Transformer encoders to extract vector representations from image-text pairs. The text encoder operates on a subword-level tokenizer. (right) The proposed bag of subwords classification only uses the single Transformer encoder.

bag-of-words and other hand-crafted rules to convert text into semi-labels. Some common practices include filtering, word segmentation, lemmatization, and the removal of numbers and stopwords to create a unique vocabulary of clean words. We found the preprocessing often eliminates long-tailed words or stopwords that contain valuable information for representation learning (see Sec. 4.4). In contrast, SuperClass simply utilizes raw word tokens as supervision signals without requiring any hand-crafted preprocessing: no filtering or removal of stopwords. Hence SuperClass preserves all information from the original text descriptions as supervision signal.

We proposed a **Super**-simple-**Class**ification approach (**SuperClass**) that simply trains to classify raw text tokens and scales as good as CLIP. As shown in Figure 1, similar to CLIP, SuperClass directly operate on text tokens with any manual text filtering. Our comprehensive empirical study shows that even without the need for a text encoder, classification methods can achieve performance comparable to the contrastive approach in terms of both model capabilities and data scalability. We demonstrate that SuperClass is a competitive alternative to its contrastive counterpart on both image classification and vision & language tasks. Pretrained on the same Datacomp-1B [21] datasets with an equal number of seen samples, SuperClass dominantly outperforms its contrastive counterparts across various of vision only and vision & language scenarios. We further explore the scaling behavior of SuperClass concerning model size and number of seen samples. Experiments suggest that classification-based methods can exhibit competitive or even superior scaling behavior compared to their contrastive counterparts. We hope our method, experiments and analysis can encourage future potentials of classification-based methods as the foundational vision-language pretraining methods.

## 2 Related Work

With the growing availability of large-scale, web-sourced image-text datasets [57, 65, 6, 68, 21, 63, 4, 62], new methods have emerged to leverage this data as supervision for training deep representations. These approaches typically involve one of three strategies: using text as classification labels, implementing image-text contrastive learning, or treating text as autoregressive targets.

**Text as classification labels.** The exploration of image-text data for model training has deep roots, with early work like Image-to-Word[54] over two decades ago aiming to enhance content-based image retrieval. This study pioneered efforts to train models to predict nouns and adjectives in text documents linked to images. Building on these early ideas, subsequent research has sought to improve data efficiency[68, 43], model effectiveness [31, 27], and vocabulary expansion [27, 78, 51, 39]. With the recent develop of network architecture, Tag2Text [27] and RAM [78] have employed Vision Transformers (ViT) [18] as vision backbones, extracting nouns from the CC12M dataset [6] and, through a combination of rules and manual selecting, arrived at 6,449 words to use as classification categories. Similarly, CatLIP [51] has filtered out "gold labels" from the CC3M [65] and Datacomp-1B [21] datasets based on certain rules, and then trained visual models using even larger-scale image-text pair datasets.

Unlike previous classification methods that rely on complex rules or manual filtering to curate "gold labels" for classification vocabularies, our approach eliminates the need for such filtering. Instead, we directly leverage text tokens as classification categories, preserving valuable textual information that might otherwise be discarded.

**Image-text contrastive learning.** Large-scale contrastive vision-language pretraining gained traction with the introduction of CLIP [57] and ALIGN [30]. Since then, numerous approaches have focused on enhancing CLIP's performance [76, 45, 44, 11, 7, 74]. For instance, SigLIP [76] reduces the computational load of CLIP's softmax-based contrastive loss by employing a sigmoid loss for local pairwise similarity calculations. LiT [77] adopts pretrained vision and language backbones with contrastive training, while other methods [45, 44, 19] aim to enhance training efficiency in image-text pretraining. InternVL [11] further innovates by integrating a large language model as the text encoder within CLIP.

In our approach, we challenge the necessity of an additional backbone to encode text information for contrastive learning. Instead, we directly use text token input as the supervisory signal, eliminating the need for text encoding and avoiding the computational overhead of large contrastive operations. This streamlined setup achieves performance comparable to dual-backbone methods.

**Text as autoregressive targets.** Various recent studies [15, 61, 38, 32, 41, 26] have delved into employing image captioning for model pretraining. SimVLM [71] has innovated this field by pioneering the pretraining of a multimodal encoder-decoder that fuses vision and language at an early stage, leveraging a hybrid architecture for applications such as visual question answering (VQA). CapPa [70] demonstrates that a simple encoder-decoder setup can efficiently pretrain vision encoders solely through captioning. Furthermore, recently studies [75, 42, 37] combine contrastive learning with captioning objectives, occasionally incorporating an additional text encoder.

In this work, we revisit the classification-based approach using large-scale visual-language datasets. Unlike the image captioning methods mentioned earlier, our classification method integrates the text captioning decoder within the vision encoder, allowing a single vision encoder to connect both modalities. Experiments demonstrate that SuperClass achieves competitive, and often superior, performance across various downstream tasks.

## 3 A simple classification approach to pretrain vision encoders

In this section, we present our proposed approach, SuperClass, which employs a classification-based pretraining method using text supervision. We begin by outlining the general framework of SuperClass. Next, we explain how text is converted into category labels without the need to select "gold labels", allowing all text to supervise the training of the image encoder. Finally, we illustrate our choice of loss design among various classification losses. Additionally, recognizing the differing significance and discriminative power of each word, we incorporated inverse document frequency as class weights in the loss design.

**Overview.** We aim to establish a pretraining method based on image classification that matches CLIP in simplicity, scalability, and efficiency. To achieve this, we follow the standard protocol by utilizing Vision Transformer (ViT) backbones as vision encoders, followed by a global average pooling layer and a linear layer as the classification head to output the logit vector $\mathbf{x}$. The supervision targets are derived from the text associated with the image, and the classification loss is computed using the text-derived classification labels and the predicted logits.

**Texts as Labels.** We directly use tokenized text as K-hot labels, where K is the number of tokens in the given sentences. More specifically, for a given image-text dataset $\mathcal{D} = \{(I_i, T_i) \mid i \in [1, N]\}$ with $N$ pairs of images $I$ and text captions $T$, we differ from previous classification-based methods by directly using an existing subword-level tokenizer, such as the one used in CLIP or BERT, with a vocabulary size $V$. This tokenizer inputs the text $T$ and obtain the set $\mathbb{C}$ of corresponding subword IDs, which serves as the classification labels. The label in the set $\mathbb{C}$ satisfies $\{c \in [1, V]\}$. The classification labels $\mathbb{C}$ will be converted into K-hot vector $\mathbf{y} \in \mathbb{R}^V$, where $y_c = 1$ when $c$ in the set $\mathbb{C}$, otherwise $y_c = 0$.

Compared to previous methods, our approach does not require any preprocessing or manual threshold setting, making it straightforward. At the same time, it also avoids the out-of-vocabulary issue that might be encountered by previous approaches.

**Classification Loss.** A significant body of research has focused on multi-label classification loss. However, it is important to emphasize that our primary goal is to pretrain vision encoders rather than prioritize multi-label classification accuracy. In a multi-label scenario, a Softmax loss is applied in SuperClass by representing labels in a probabilistic manner, where $\hat{y}_c$ is a normalized weighted label.

$$\ell_{ce} = -\sum_{c=1}^{V} \hat{y}_c \log \frac{e^{x_c}}{\sum_{c'} e^{x_{c'}}} \tag{1}$$

We evaluated several multi-label classification losses, including Softmax loss, BCE loss, soft margin loss, ASL [59], and two-way loss [33]. Surprisingly, the simple Softmax loss yielded the best pretraining results. This may be due to the fact that existing multi-label classification losses assume that labels are precise and exhaustive, aiming to optimize the margin between positive and negative classes. However, the inherent noise in image-text data and the limitations of text in fully capturing an image's content mean that not all objects in an image are always referenced in the accompanying text.

**Inverse Document Frequency.** Within the subword vocabulary, not all categories contribute semantically equally, as different words carry varying amounts of information. Additionally, the subword dictionary contains many words unrelated to visual content that frequently appear in sentences, which do not provide effective supervisory information. Therefore, words with higher information content should be given greater weight during training. To achieve this, we employ Inverse Document Frequency (IDF) as a measure of information significance. The fewer the number of samples containing a specific word, the stronger its ability to differentiate between samples. We use the IDF statistic of each category (subword) as the weight for the corresponding classification label, assigning different weights $w_c$ to the classification labels $c$.

$$\hat{y}_c = \frac{w_c y_c}{\sum_{c'} w_{c'} y_{c'}}, \quad w_c = \log \left( \frac{|\mathcal{D}|}{1 + \mathrm{df}(c)} \right) \tag{2}$$

where $|\mathcal{D}|$ denotes the total number of image-text pairs, $\mathrm{df}(c)$ is the document frequency (df) of subword $c$, in other words, it's the count of texts containing subword $c$. For greater ease of use, we have implemented an online IDF statistic that is computed during the training process, eliminating the need for pre-training offline statistics. This approach enhances the user-friendliness and portability of our method.

## 4 Experiment

### 4.1 Experiment setup

We use a standard subset of the datacomp dataset [21] for pre-training, which contains about 1.3B image-text pairs. A batch size of 16k and 90k are adopted for our classification models and CLIP models. In the ablation section, all experiments are conducted with a batch size of 16k. To make a fair comparsion with the CLIP, we use 90k batch size, adopt the AdamW with a cosine schedule, and set the same learning rate and decay as CLIP.

### 4.2 Evaluation protocols

In order to better highlight the effectiveness of pretraining method, we concentrate on the properties of the frozen representations.

**Linear probing** We evaluate the classification accuracy when using the full ImageNet-1k [14] training set to learn a dense projection layer and frozen the parameters of backbone. We follow the linear probing training recipe from MAE [24].

Table 1: Comparison of the Linear probing top-1 accuracy on ImageNet-1K dataset.

| Method | PreTraining data | ViT-Base | | ViT-Large | |
|--------|------------------|----------|--|-----------|--|
| | | #Seen Samples | Top-1 (%) | #Seen Samples | Top-1 (%) |
| *contrastive or clustering based* | | | | | |
| MoCov3 [10] | IN1K | 400M | 76.7 | 400M | 77.6 |
| DINO [5] | IN1K | 512M | 78.2 | - | - |
| iBOT [80] | IN22K | 400M | 79.5 | 256M | 81.0 |
| DINOv2 [55] | LVD-142M | - | - | 2B | 84.5 |
| *reconstruction based* | | | | | |
| BEiT [3] | D250M+IN22K | 1B | 56.7 | 1B | 73.5 |
| SimMIM [73] | IN1K | 1B | 56.7 | - | - |
| CAE [8] | D250M | 2B | 70.4 | 2B | 78.1 |
| MAE [24] | IN1K | 2B | 68.0 | 2B | 75.8 |
| *vision-language pretraining based* | | | | | |
| Openai CLIP [57] | WIT-400M | 13B | 78.5 | 13B | 82.7 |
| Cappa [70] | WebLI-1B | - | - | 9B | 83.0 |
| OpenCLIP [29] | Datacomp-1B | - | - | 13B | 83.9 |
| SuperClass | Datacomp-1B | 1B | 78.7 | 1B | 82.6 |
| SuperClass | Datacomp-1B | 13B | **80.2** | 13B | **85.0** |

Table 2: Performance of frozen visual representations on different classification datasets. 10-shot linear evaluation accuracy on the pre-logit representation. *results from the paper.

| Method | ImageNet | Pets | Cars |
|--------|----------|------|------|
| MAE | $44.0_{\pm 0.1}$ | $57.7_{\pm 0.2}$ | $32.5_{\pm 0.1}$ |
| DINOv2 | $77.0_{\pm 0.1}$ | $94.2_{\pm 0.1}$ | $76.8_{\pm 0.2}$ |
| CapPa* | $70.6_{\pm 0.2}$ | $92.6_{\pm 0.5}$ | $92.2_{\pm 0.2}$ |
| OpenCLIP | $75.6_{\pm 0.1}$ | $92.2_{\pm 0.6}$ | $\mathbf{92.7_{\pm 0.3}}$ |
| SuperClass | $\mathbf{77.2_{\pm 0.1}}$ | $\mathbf{94.6_{\pm 0.1}}$ | $92.6_{\pm 0.1}$ |

Table 3: Zero-shot Top-1 acc. and CIDEr are tested on ImageNet-1k dataset and COCO captions, respectively. The zero-shot accuracy of SuperClass is obtained after lock-image tuning [77].

| Case | Backbone | 0-shot | CIDEr |
|------|----------|--------|-------|
| Openai CLIP | ViT-L/14 | 75.3 | **113.5** |
| OpenCLIP | ViT-L/14 | 79.2 | - |
| CLIP,reimpl. | ViT-L/16 | 79.0 | 112.6 |
| SuperClass | ViT-L/16 | **79.7** | 113.0 |

**10-shot classification** Following the setting of Cappa [70], we perform 10-shot classification on ImageNet-1k [14], Pets [56] and Cars [34]. For each dataset and model, we run 3 times, and report the mean results and variance.

**Locked-image Tuning** Locked-image Tuning (LiT) [77] employs contrastive training to align locked image and unlocked text models. Generally, LiT is an efficient way to equip any pretrained vision backbone with zero-shot classification and retrieval capabilities. We follow the setup from LiT[77] and assess the zero-shot classification accuracy on ImageNet-1k [14].

**Collaborating with language models** Motivated by recent works [47, 11, 1, 67, 2, 71] combining pretrained vision backbones [57, 19, 76] and language models [69, 13], we investigate the amenability of the learned representations to interface with a text decoder. Here, we evaluate two ways to collaborate with large language models. 1) following ClipCap [53], we frozen both pretrained image encoder and pretrained 12-layer GPT2 decoder [58], only train an adapter to connect the image encoder and language model to perform image captioning on COCO captions [9]. 2) following LLaVA [47] setup, we train and finetune a projection layer and a pretrained large language models, Vicuna-7B [13] to solve downstream tasks, including VQAv2(val) [22], GQA [28], VizWiz(val) [23], T-VQA(val) [66], SQA(img) [79], MMBench(en) [48], MME [20], POPE [46], MMMU [25] and SEEDBench [40].

## 4.3 Main results

**Comparison with different types of pretraining methods** In Table 1, we compare the models trained by SuperClass with the different types of pretraining methods, including contrastive or clustering based methods [10, 5, 80, 55], reconstruction based [3, 73, 8, 24], and vision-language

pretraining based methods [57, 70, 21]. In general, the proposed method achieves best performance among these pretraining methods.

Compared to the current SOTA self-supervised model DINOv2 [55], our method achieves a 0.5% higher accuracy in IN-1K linear probing (85.0 vs 84.5) without a bunch of bells and whistles. Although SuperClass has seen more samples, it operates as a simpler classification framework and does not employ MultiCrop, Masked Image Modeling, or Contrastive learning, as DINOv2 does. Although comparing a self-supervised learning method to a (weakly) supervised learning approach is a system-level comparison, we still observe that a simple classification pretraining approach demonstrates superior performance across many classy benchmarks that shown in Table 2.

Compared to the contrastive counterparts CLIP [57], our method achieves higher linear probing top-1 accuracy on ImageNet-1K dataset with ViT-Base (80.2 vs 78.5) and ViT-Large (85.0 vs 82.7) as backbone. For a fair comparison, we further make a comparison with OpenCLIP [29] which trains a ViT-Large model with a batch size 90k based on Datacomp-1B dataset. Our method consistently outperforms OpenCLIP by a large margin (85.0 vs 83.9). In Table 2, our method surpasses OpenCLIP on IN-1K and Pets by clear margins with improvements of 1.6 and 2.2 points, while being comparable with OpenCLIP on Cars (92.6 v.s 92.7).

**Further comparison with CLIP**    In Table 3, we compare the models trained by SuperClass with the currently widely used CLIP models, including zero-shot classification, and COCO captioning. To verify the effectiveness of the pretraining method, we adapt the standard ViT [18] structure as the visual backbone and added a classification head on top of it. We use the open-source Datacomp-1B [21] dataset and encounter 13B samples in the training process.

For a better comparison with the CLIP models, we select three types: OpenAI's CLIP ViT-L/14 trained on the internal WiT-400M data, and Laion's CLIP ViT-L/14 trained on the open-source Datacom-1B dataset. The checkpoints of these two models have been open-sourced. The checkpint of Laion's openCLIP is downloaded from Hugginface Hub[1]. For a fair comparison, we trained the ViT-L/16 model with a batch size 90k based on the our codebase and Datacomp-1B dataset.

With Lock image Tuning [77], the trained classification model also gains the ability of zero-shot classification. Our method achieves 79.7% Top-1 zero-shot accuracy on ImageNet-1k datatset which is much better than OpenAI CLIP ViT-L/14 and OpenCLIP ViT-L/14. Although maybe they are not directly comparable, this do reflect that the vision model trained by the proposed SuperClass is with strong visual perception capabilities.

Combining the frozen vision encoder with a frozen pretrained 12-layer GPT-2 decoder [58] via a trained adapter, the models are trained on COCO captions [9] and CIDEr socres are reported. We observe that the CIDEr socres of our method are slightly below OpenAI's CLIP, which may be due to the use of different datasets; OpenAI's CLIP utilizes an internal dataset, WiT-400M. However, our approach outperforms our implemented CLIP model with the same settings.

Overall, the models trained by proposed SuperClass demonstrated marginally improved accuracy in both classification capabilities and the vision & language task when compared to the contrastive pretrained CLIP models.

**Large multi-modal models**    Many large multi-modal models integrate pre-trained vision backbones with large language models. We explore how amenable the learned representations are to interfacing with a text decoder. Following the LLaVA setup [47], we combine frozen CLIP models and SuperClass models with the pretrained Vicuna-7B [13] and perform downstream tasks. In Figure 2, we show some results of vision&language downstream tasks. The results demonstrate that SuperClass models could achieve better performance than CLIP models on the majority of datasets. It is worth mentioning that, in comparison to CLIP models, SuperClass models exhibit significantly better performance on VQAv2 [22], T-VQA [66], and MMBench [48], which pertain to OCR and fine-grained recognition tasks, respectively. In addition, the overall accuracy measurement on VizWiz [23] are not stable due to a significant portion of questions being labeled as unanswerable. To ensure the completeness of our findings, we still present the results on this dataset.

Due to space limitations, detailed numerical results are provided in the appendix. Additionally, you can find more experimental results in the appendix.

---

[1]Laion's CLIP https://huggingface.co/laion/CLIP-ViT-L-14-DataComp.XL-s13B-b90K.

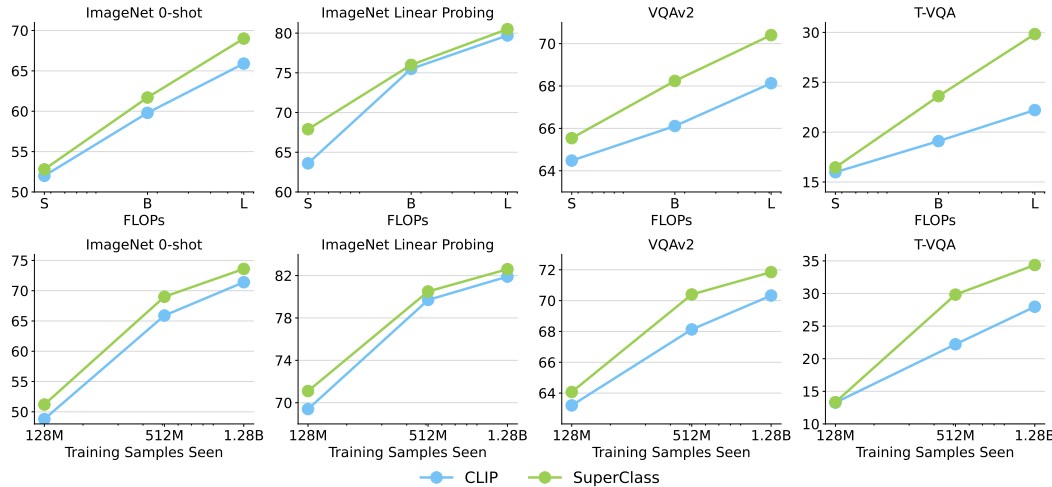

Figure 2: Zero-shot classification accuracy and linear probing accuracy on ImageNet-1k dataset (left two columns); Performance of VQAv2 and T-VQA with LLaVA training recipe (right two columns). **Top row:** We compare the performance of vision backbones—ViT-S/16, B/16, and L/16—pretrained via classification and contrastive methods with the same batch size of 16k and 512 million seen samples, focusing on their size and computational cost. SuperClass demonstrates better scaling on zero-shot classification and VQAv2, T-VQA tasks. **Bottom row:** Comparing SuperClass and CLIP, performance increases with more training examples, mirroring the effects of model scaling. All methods are trained the same batch size of 16k and ViT-L/16 as backbone.

**Model scaling results**    In the top row of Figure 2, we showcase the performance across classification and vision & language tasks for varying model scales. For a fair comparison, both CLIP and SuperClass models undergo training with identical settings, which include a batch size of 16k and 512 million seen samples. As shown in Figure 2, with the model scaling up, we observe a corresponding enhancement in performance, whether it is for classification tasks or the downstream tasks associated with LLaVA. Generally speaking, with the same model size, models pre-trained using SuperClass exhibit superior precision compared to those trained with CLIP. SuperClass demonstrates better scaling on zero-shot classification and VQAv2, T-VQA tasks.

**Data Scaling results**    In the bottom row of Figure 2, we showcase the performance across classification and vision & language tasks for varying seen samples. For a fair comparison, both CLIP and SuperClass models undergo training with identical settings, which include a batch size of 16k and ViT-L/16 as backbone. Figure 2 illustrates that as the number of seen samples grows, there is a noticeable improvement in performance for both classification and downstream tasks linked to LLaVA. Typically, models pre-trained with SuperClass outperform those trained with CLIP in terms of accuracy when given the same amount of seen samples. SuperClass exhibits the same or slightly better scaling behavior compared to CLIP on downstream tasks. In addition, SuperClass does not require a text encoder, it offers better efficiency in training compared to CLIP.

### 4.4 Ablations

To verify the rationality of the SuperClass, we conduct extensive ablation experiments that pretrain on datacomp-1B [21] and evaluate on several downstream tasks with different settings for SuperClass.

**Word-level tokenizer vs. Subword-level tokenizer**    Table 4 presents the results of word-level tokenizer and subword-level tokenizer on serval daownstream tasks. We use the tokeinzer in openai CLIP as our subword-level tokenizer. We compare it with the word-level tokenizer used in CatLIP [51], which carefully selected approximately 40,000 "gold labels" from the datacomp-1B dataset. Aside from the tokenizer being different, all models are trained under the same settings.

For ViT-S/16, word-level tokenizer achieves better classification accuracy than subword-level tokenizer. A possible reason is that when the model capacity is limited, the filtered clean supervisory

Table 4: Word tokenizer vs. Subword tokenizer. The performance of classification and LLaVA downstream tasks with different tokenizers. SuperClass use a subword-level tokenizer to map text into category labels. All models are trained in the same settings with a batch size 16k and 512M seen samples.

| Tokenizer | ViT | Classification | | Vision & Language Downstream Tasks | | | | | | | | | |
|---|---|---|---|---|---|---|---|---|---|---|---|---|---|
| | | LP | ZS | VQAv2 | GQA | VizWiz | T-VQA | SQA | MMBench | MME | POPE | MMMU | SEED |
| Word | S/16 | **68.4** | **53.2** | 65.28 | 55.38 | 43.12 | 15.84 | **65.83** | **49.14** | 1228 | 80.32 | **36.6** | **50.54** |
| Subword | S/16 | 67.9 | 52.8 | **65.54** | **55.95** | **46.23** | **16.46** | 65.64 | 48.53 | **1306** | **81.02** | 33.2 | 50.43 |
| Word | B/16 | **76.1** | 61.4 | 67.72 | 57.12 | **46.20** | 20.98 | **65.79** | **54.63** | 1296 | 81.88 | 34.4 | **53.38** |
| Subword | B/16 | 76.0 | **61.7** | **68.34** | **57.43** | 41.79 | **24.41** | 65.54 | 54.03 | **1324** | **82.28** | **36.9** | 52.88 |
| Word | L/16 | 80.3 | 68.2 | 69.47 | 57.36 | **51.94** | 23.87 | 65.00 | 56.27 | 1335 | 83.28 | 35.4 | 53.64 |
| Subword | L/16 | **80.5** | **69.0** | **70.40** | **58.16** | 51.48 | **29.83** | **67.72** | **59.45** | **1373** | **84.04** | **36.3** | **53.74** |

Table 5: The performance of classification and LLaVA downstream tasks with different subword-level tokenizers. All models are trained in the same settings with a batch size 16k, 512M seen samples and ViT-L/16 as Backbone.

| Tokenizer | Vocab | Classification | | Vision & Language Downstream Tasks | | | | | | | | | |
|---|---|---|---|---|---|---|---|---|---|---|---|---|---|
| | | LP | ZS | VQAv2 | GQA | VizWiz | T-VQA | SQA | MMBench | MME | POPE | MMMU | SEED |
| OpenaiCLIP | 49,152 | **80.5** | **69.0** | **70.40** | **58.16** | **51.48** | **29.83** | **67.72** | **59.45** | 1373 | **84.04** | **36.3** | 53.74 |
| WordPiece | 32,000 | **80.5** | 68.5 | 69.95 | 57.76 | 49.07 | 29.33 | 65.99 | 56.18 | **1375** | 83.37 | 35.1 | **54.05** |
| SentencePiece | 32,000 | 80.2 | 67.8 | 69.52 | 57.95 | 49.20 | 28.56 | 65.05 | 57.47 | 1301 | 82.16 | 34.8 | 53.52 |

information may be more conducive to model convergence. However, with the increasing size of the model, subword-level tokenizer gradually outperforms the word-level tokenizer, whether in classification tasks or vision & language tasks.

Regardless of model size, on most vision & language tasks, subword-level tokenizer tends to perform better than word-level tokenizer. The reason may be that subword-level tokenizer retain a substantial amount of language-related information, although it may not be highly relevant to visual information. This makes the features of the models trained with the subword-level tokenizer more readily integrated with large language models.

Overall, using a subword-level tokenizer exhibits better scaling behavior and is more suitable for use in large multi-modal models.

**Different subword-level tokenizers**   Table 5 presents the results on classification tasks and LLaVA downstream tasks with different subword-level tokenizers. Here, we compare the character-based byte pair encoding tokenizer [64] used in CLIP [57], the WordPiece [72] tokenizer used in BERT [16] and the SentencePiece [36] tokenizer used in LLama [69], they are all subword-level tokenizers. The tokenizer used in openai CLIP obtains best performance on the classification task and LLaVA downstream tasks. Finally, we choose the tokenizer used in CLIP [57] for the training of SuperClass models.

**Classification loss**   Table 6 represents different classification loss on ImageNet-1k dataset. We selected several of the most commonly used multi-label classification losses for experimentation. **Softmax loss** is often used in single-label classification tasks. It is possible apply a softmax loss in a multi-label scenario through describing labels in a probabilistic way. **BCE loss** is a binary cross-entropy (BCE) loss and is often used in multi-label classification tasks as baseline. Asymmetric Loss(**ASL loss**) a improved BCE loss to address positive-negative imbalance. **Soft margin loss** is a

Table 6: The performance on classification tasks with different classification losses. All models are trained in the same settings with a batch size 16k, 512M seen samples and ViT-B/16 as Backbone.

| Loss&Acc. | Softmax | BCE | ASL | SoftMargin | Two-way |
|---|---|---|---|---|---|
| Linear prob | **75.6** | 73.6 | 73.8 | 73.5 | 74.8 |
| Zero-shot | **60.8** | 58.5 | 58.7 | 58.1 | 59.7 |

Table 7: The effect of IDF weight in the loss and removing stopwords.

| Tokenizer | Classification | | Vision & Language Downstream Tasks | | | | | | | | | |
|---|---|---|---|---|---|---|---|---|---|---|---|---|
| | LP | ZS | VQAv2 | GQA | VizWiz | T-VQA | SQA | MMBench | MME | POPE | MMMU | SEED |
| SuperClass | **76.0** | **61.7** | **68.34** | **57.43** | 41.79 | **24.41** | **65.54** | 54.03 | 1324 | 82.28 | **36.9** | 52.88 |
| w/o IDF | 75.6 | 60.8 | 68.08 | 57.27 | 47.60 | 23.73 | 65.44 | **54.55** | 1310 | **82.58** | 34.6 | 52.53 |
| rm Stopwords | 75.7 | 61.0 | 68.29 | 57.41 | **47.67** | 24.12 | 65.34 | 53.86 | **1343** | 82.50 | 33.8 | **53.12** |

margin-based loss for multi-label classification tasks. **Two-way loss** is the current state-of-the-art (SOTA) for multi-label classification tasks.

Surprisingly, the simplest softmax loss outperforms all other multi-label classification losses by a large margin. We believe that existing multi-label classification losses operate under the assumption that labels are both accurate and complete, aiming to optimize the classification margin between positive and negative classes. However, in reality, image-text data contains considerable noise, and a single text passage cannot possibly capture all the contents of an image. Consequently, certain categorical objects present in the image may not be mentioned in the associated text. In the context of image-text pretraining, how to design a better loss function remains a question worthy of exploration.

**IDF as class weights** Considering that the importance of each category (subword) in the vocabulary is not equal and the information they carry varies, we use IDF as class weights. The Table 7 repsents the results of with and without IDF as class weights. SuperClass without IDF experienced a noticeable decrease in accuracy on classification tasks, the change in precision on LLaVA tasks is not significant.

**Removing stopwords?** Stop words are commonly used in Text Mining and Natural Language Processing (NLP) to eliminate words that are so widely used that they carry very little useful information. In the previous classification methods, the stopwords are removed. The stopwords are download from NLTK [49]. However, the results in Table 7 shows that the keeping stopwords could help the vision encoder to gain better performance on classification tasks.

## 5 Limitation and Conclusion

We have conducted a thorough comparison of vision encoders pre-trained with contrastive and classification objectives and determined that models pre-trained with classification surpass CLIP models in both classification and vision & language tasks. Additionally, our approach does not require a text encoder, which leads to higher training efficiency than that of CLIP. Furthermore, our findings suggest that classification as a pre-training task may have beneficial scaling properties as model and data sizes increase, and we encourage future research to delve into this possibility.

While it delivers impressive results on various downstream tasks, it completely ignore word order and object relationships, which implies that we are losing important supervisory information. Addressing this will be the direction of our future research efforts.

To sum up, we have demonstrated that straightforward image classification can serve as an effective pre-training strategy for vision backbones derived from image-text data. Our aim is to stimulate subsequent studies to pay more attention to the benefits of classification as a pre-training task for vision encoders.

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

Table 8: The performance of classification and LLaVA downstream tasks with different seen samples. "LP" means linear probing and "ZS" means zero-shot classification, these two are tested on ImageNet-1K dataset. The results of vision&language downstream tasks are obtained by combine the frozen vision models and Vicuna-7B [13], following the settings in LLaVA [47].

| Method | Data | Classification | | Vision & Language Downstream Tasks | | | | | | | | | |
| | | LP | ZS | VQAv2 | GQA | VizWiz | T-VQA | SQA | MMBench | MME | POPE | MMMU | SEED |
| --- | --- | --- | --- | --- | --- | --- | --- | --- | --- | --- | --- | --- | --- |
| CLIP | 128M | 69.4 | 48.8 | 63.20 | 53.98 | 45.80 | 13.27 | 64.70 | 46.56 | 1216 | 78.20 | 35.1 | 47.84 |
| SuperClass | 128M | **71.1** | **51.2** | **64.07** | **54.96** | **49.21** | **13.33** | **65.44** | **49.14** | **1241** | **80.01** | **35.3** | **49.08** |
| CLIP | 512M | 79.7 | 65.9 | 68.13 | 57.32 | 44.92 | 22.21 | 64.45 | 51.54 | 1299 | 82.48 | 35.3 | 53.06 |
| SuperClass | 512M | **80.5** | **69.0** | **70.40** | **58.16** | **51.48** | **29.83** | **67.72** | **59.45** | **1373** | **84.04** | **36.3** | **53.74** |
| CLIP | 1.28B | 81.9 | 71.4 | 70.33 | 58.95 | 46.71 | 27.97 | 64.65 | 55.49 | 1351 | 83.37 | 35.7 | 55.09 |
| SuperClass | 1.28B | **82.6** | **73.6** | **71.85** | **59.09** | **51.70** | **34.37** | **65.94** | **59.70** | **1392** | **84.41** | **36.8** | **55.51** |

Table 9: The performance of classification and LLaVA downstream tasks with different model sizes.

| Method | ViT | Classification | | Vision & Language Downstream Tasks | | | | | | | | | |
| | | LP | ZS | VQAv2 | GQA | VizWiz | T-VQA | SQA | MMBench | MME | POPE | MMMU | SEED |
| --- | --- | --- | --- | --- | --- | --- | --- | --- | --- | --- | --- | --- | --- |
| CLIP | S/16 | 63.6 | 52.0 | 64.48 | 55.26 | 44.16 | 15.98 | **65.84** | 47.33 | 1227 | 80.49 | **35.8** | 49.72 |
| SuperClass | S/16 | **67.9** | **52.8** | **65.54** | **55.95** | **46.23** | **16.46** | 65.64 | **48.53** | **1306** | **81.02** | 33.2 | **50.43** |
| CLIP | B/16 | 75.5 | 59.8 | 66.11 | 56.28 | **49.17** | 19.10 | 64.30 | 48.62 | 1289 | 81.47 | 35.9 | 50.76 |
| SuperClass | B/16 | **76.0** | **61.7** | **68.34** | **57.43** | 41.79 | **24.41** | **65.54** | **54.03** | **1324** | **82.28** | **36.9** | **52.88** |
| CLIP | L/16 | 79.7 | 65.9 | 68.13 | 57.32 | 44.92 | 22.21 | 64.45 | 51.54 | 1299 | 82.48 | 35.3 | 53.06 |
| SuperClass | L/16 | **80.5** | **69.0** | **70.40** | **58.16** | **51.48** | **29.83** | **67.72** | **59.45** | **1373** | **84.04** | **36.3** | **53.74** |

# A    Appendix / supplemental material

## Broader Impacts

This work presents a new approach to train vision models, which can be applied for image recognition and other vision tasks. The approach demonstrates higher efficiency than the popular one in the community, which can reduce the computational cost and the power cost for computer vision model training.

**Data Scaling results**    In Table 8, we showcase the performance across classification and vision & language tasks for varying seen samples. For a fair comparison, both CLIP and SuperClass models undergo training with identical settings, which include a batch size of 16k and ViT-L/16 as backbone.

Figure 2 illustrates that as the number of seen samples grows, there is a noticeable improvement in performance for both classification and downstream tasks linked to LLaVA. Typically, models pre-trained with SuperClass outperform those trained with CLIP in terms of accuracy when given the same amount of seen samples. SuperClass exhibits the same or slightly better scaling behavior compared to CLIP on downstream tasks. In addition, SuperClass does not require a text encoder, it offers better efficiency in training compared to CLIP.

**Model scaling results**    In Table 9, we showcase the performance across classification and vision & language tasks for varying model scales. For a fair comparison, both CLIP and SuperClass models undergo training with identical settings, which include a batch size of 16k and 512 million seen samples.

Table 10: Performance of frozen visual representations trained via image classification (SuperClass) and constrastively (CLIP). Linear probing and zero-shot classification are both tested on ImageNet-1k dataset. Captioning is conducted on COCO captions and CIDEr is reported in the table. The zero-shot accuracy of SuperClass is obtained after lock-image tuning [77].

| Method | Backbone | Data | Seen Samples | Zero-shot | Linear Probing |
|---|---|---|---|---|---|
| CLIP | RN-50 | Datacomp-1B | 1.28B | 60.73 | 70.28 |
| SuperClass | RN-50 | Datacomp-1B | 1.28B | 62.81 | 71.92 |
| CLIP | ConvNext-tiny | Datacomp-1B | 1.28B | 59.94 | 70.35 |
| SuperClass | ConvNext-tiny | Datacomp-1B | 1.28B | 62.85 | 72.33 |

Table 11: The performance of vision & language downstream tasks with different pretrained models.

| Method | VQAv2 | GQA | VizWiz | T-VQA | SciQA | MME | MMB | PoPE | MMMU |
|---|---|---|---|---|---|---|---|---|---|
| OpenCLIP | 74.54 | 61.03 | 50.47 | 38.16 | **67.33** | **1434**/269 | 60.73 | 85.52 | 35.9 |
| MAE | 63.50 | 54.58 | 50.22 | 11.55 | 54.75 | 1175/343 | 42.44 | 80.69 | 35.7 |
| DINOv2 | 73.32 | **61.87** | 49.15 | 14.08 | 64.90 | 1336/297 | 57.90 | **86.24** | 35.3 |
| SuperClass | **75.24** | 60.96 | **54.33** | **39.20** | 66.09 | 1371/**322** | **63.14** | 85.69 | **36.0** |

As shown in Figure 2, with the model scaling up, we observe a corresponding enhancement in performance, whether it is for classification tasks or the downstream tasks associated with LLaVA. Generally speaking, with the same model size, models pre-trained using SuperClass exhibit superior precision compared to those trained with CLIP. SuperClass demonstrates better scaling on zero-shot classification and VQAv2, T-VQA tasks.

**Superclass with different model types**  To evaluate the robustness of our proposed method across different model types, we selected two representative convolution-based networks: ResNet50 and ConvNext-Tiny. We compare SuperClass against CLIP for ImageNet zero-shot (LiT) and linear probing classification, as shown in Table 10. All experiments were conducted with a batch size of 16k and 1.28B seen samples. We observe that SuperClass surpasses CLIP in all settings by a clear margin, ranging from 1.64 to 2.91. These results demonstrate that the superiority of SuperClass over CLIP is robust across different model architectures.

**VLM downstream tasks with different types of pretraining models**  Following the LLaVA setup, we combine frozen CLIP models, self-supervised models, and SuperClass models with the pre-trained Vicuna-7B and perform downstream tasks. The experimental results in Table 11 demonstrate that the proposed method could achieve better than self-supervised ViT pre-training methods, like DINOv2, and weakly-supervised methods, like CLIP.

**Comparison with other classification based pretraining models**  We have included the comparison with other classification-based methods, like CatLIP [51] in the subsection Word-level tokenizer vs. Subword-level tokenizer. The word-level tokenizer is used in CatLIP [51], which carefully selected approximately 40,000 "gold labels" from the datacomp-1B dataset. Aside from the tokenizer being different, all models are trained under the same settings. The results of Table 4 show that with the increasing size of the model, the subword-level tokenizer gradually outperforms the word-level tokenizer, whether in classification tasks or vision & language tasks. We also provide the results of finetuning on ImageNet-1k in Table 12. Using the same dataset Datacom-1B for training, the same backbone ViT-L/16 as backbone, the same number 13 Billion of training samples seen, the SuperClass could achieve better performance than CatLIP (87.8 vs 86.5).

Table 12: Comparison of the Fine-tuning top-1 accuracy on ImageNet-1K dataset. *number from the paper.

| Method | Pretraining Data | ImageNet-1k Fine-tuning |
|---|---|---|
| OpenCLIP ViT-L/14 | Datacomp-1B | 87.4 |
| CatLIP ViT-L/16* | Datacomp-1B | 86.5 |
| Superclass ViT-L/16 | Datacomp-1B | 87.8 |

