# OpenReview forum: "Classification Done Right for Vision-Language Pre-Training"
_NeurIPS.cc/2024/Conference — NeurIPS 2024 poster_

### Official Review · Reviewer_bBvT · 2024-06-27

**Soundness:** 4
**Presentation:** 4
**Contribution:** 3
**Rating:** 7
**Confidence:** 5

**Summary:**

The paper proposes a simple alternative to CLIP-style pretraining that doesn't require a text encoder and can be done using only a text tokenizer. The goal is to provide a simpler yet more efficient alternative to vision-language model (VLM) pretraining, which is known to be very expensive. Additionally, the authors simplify the VLM pertaining setup into a classification task which is both novel and very intuitive.
The authors demonstrate the efficiency of their method through experiments on both classification and vision-and-language downstream tasks. Additionally, they present a comprehensive set of ablations to dissect the performance gains of their proposed approach.

While there are some limitations to their proposed method, the simplification offers valuable benefits for traditional classification and vision-language tasks.

**Strengths:**

1. The paper is well-written and easy to follow, with clear motivation and well-planned experiments.
2. The proposed method simplifies vision-language pretraining by eliminating the need for a text encoder, using a text tokenizer instead. This approach demonstrates comparable or better performance on both classification and vision-language tasks. It effectively transforms the contrastive pretraining task into an open vocabulary classification task, where text tokens provide supervision. This insight is powerful as it can help the community develop strong vision encoders more efficiently.
3. The experiments and ablations presented are thorough. The authors systematically analyze every component of their method, including data, model scale, and the tokenizer.
4. The paper also highlights that for simple classification tasks or vision question-answering tasks, CLIP-like pretraining is not necessary for learning strong, robust vision encoders. This is a novel insight of the paper.

**Weaknesses:**

1. For classification tasks, the authors only present ImageNet-1K classification accuracy and do not perform experiments on other popular few-shot and zero-shot classification benchmarks. Including these benchmarks would have strengthened the paper.
2. Vision-language models like CLIP are also used in text-to-image generation systems. The authors do not address this aspect or present any experiments showcasing the effects of using their encoder in such systems.
3. This work shares similarities with research that cleans pretraining captions before performing CLIP-style pretraining [1]. However, the authors do not compare their method against such approaches. I believe this comparison is important, as synthetic captions are increasingly used for CLIP-style pretraining.
4. There is a small reference error in line 235.




[1] Fan, Lijie, et al. "Improving clip training with language rewrites." Advances in Neural Information Processing Systems 36 (2024).

**Questions:**

1. In Table 7, the authors show that removing stop-words has no effect and state in line 279 that "keeping stopwords could help the vision encoder." This statement is not explained and seems counter-intuitive. Could the authors elaborate on this observation?
2. The authors claim they can use IDF weights in an online manner. Could the authors explain how they achieve this? I don't understand how this process can be done online.
3. The authors demonstrate that the performance of SuperClass is as good as or better than traditional CLIP-style pretraining on ImageNet-1K classification. However, it would be interesting if the authors could specify which classes benefit more from their method compared to traditional CLIP. Such an analysis could help understand where the gains stem from.

**Limitations:**

1. The authors acknowledge that their approach completely ignores word order, which can significantly impact the encoder's ability to understand tasks requiring spatial reasoning.
2. Additionally, this approach may not be suitable for text-to-image systems, as these models need to understand word order and infer relationships from text to generate accurate images.

---

> ### Author Rebuttal · Authors · 2024-08-07
>
> > **W1**. Other popular few-shot and zero-shot classification benchmarks.
>
> **A1**. Thanks for the advice. We have added more evaluation benchmarks, including 10-shot classification on ImageNet-1k, pets and cars, and zero-shot classification on 8 more datasets.
>
> **10-shot classification**
> We follow the setting of Cappa[1]. For each dataset and model, we run 3 times, and report the mean results and variance in the following table. Our method surpasses CLIP on IN-1K and Pets by clear margins with improvements of 1.6 and 2.2 points, while being comparable with CLIP on Cars (92.6 v.s 92.7).
>
> | Case        | IN-1K         | Pets          | Cars          |
> |-------------|---------------|---------------|---------------|
> | MAE         | 44.0(0.1)     | 57.7(0.2)     | 32.5(0.1)     |
> | Dinov2      | _77.0(0.1)_   | _94.2(0.1)_   | 76.8(0.2)     |
> | Cappa*      | 70.6(0.2)     | 92.6(0.5)     | 92.2(0.2)     |
> | CLIP        | 75.6(0.1)     | 92.2(0.6)     | **92.7(0.3)** |
> | Superclass  | **77.2(0.1)** | **94.6(0.1)** | _92.6(0.1)_   |
>
> **zero-shot classification**
> Following LiT[2], we train a text encoder on Datacomp-1B by keeping the image encoder locked. In this way, we are able to perform zero-shot classification. We tested the model on the following 10 datasets. Superclass beats openCLIP on seven of the datasets. It is worth noting that SuperClass uses a ViT large model with patch size 16, while the baseline method adopts a patch size of 14. This makes the superiority of SuperClass over openCLIP even clearer.
>
> | Case       | Model    | Data        | Seen samples | IN-1K | imagenet-v2 | imagenet-r | imagenet-a | imagenet-sketch | GTSRB | Rendered SST2 | ObjectNet | SUN397 | Country211 |
> |------------|----------|-------------|--------------|-------|-------------|------------|------------|-----------------|-------|---------------|-----------|--------|------------|
> | openCLIP   | ViT-L/14 | Datacomp-1B | 12.8B        | 79.2  | 72.1        | 90.8       | 69.6       | 68.0            | 58.5  | 61.0          | 74.3      | 74.3   | 31.6       |
> | SuperClass | ViT-L/16 | Datacomp-1B | 12.8B        | 79.7  | 72.4        | 91.6       | 68.8       | 70.6            | 58.5  | 61.6          | 73.9      | 73.8   | 32.3       |
>
> [1] Tschannen, Michael, et al. "Image captioners are scalable vision learners too." NeurIPS 2024.
> [2] Zhai, Xiaohua, et al. "Lit: Zero-shot transfer with locked-image text tuning." CVPR. 2022.
>
>
> > **W2**. The application in text-to-image generation systems.
>
> **A2**. Thanks for the advice. Text-to-image generation is indeed a particularly important application of vision-language models. However, due to the limited time available for rebuttal, we were unable to attempt and train this task. We will explore this issue further in the future.
>
> > **W3**. The effect of synthetic captions
>
> **A3**. Thanks for the nice advice. We use the code provided by LaCLIP[1] https://github.com/LijieFan/LaCLIP for investigation. We compare SuperClass against CLIP with ViT-B/16 following the setting of LaCLIP. The numbers of CLIP and LaCLIP are directly borrowed from the paper.
> As shown by the results in the following table, Superclass can also benefit from rewritten captions, and the improvement is even greater than that of CLIP (+1.1 vs. +1.6 in zero-shot and +1.2 vs. 1.9 in Linear probing). The possible reason is that the rewritten captions transform the sentence structure but keep the major objects and subjects intact, indirectly enhancing the weight of those visual-related words.
>
> | Case       | Data       | epoch |  Zero-shot | Linear Probing |
> |------------|------------|-------|------------|----------------|
> | CLIP       | CC3M       | 25    | 15.8       | 54.5           |
> | SuperClass | CC3M       | 25    | 16.9(+1.1) | 55.7(+1.2)     |
> | LaCLIP     | CC3M recap | 25    | 21.5       | 56.5           |
> | SuperClass | CC3M recap | 25    | 23.1(+1.6) | 58.4(+1.9)     |
>
> [1] Fan, Lijie, et al. "Improving clip training with language rewrites." NeurIPS 2024.

---

> ### Author Response · Authors · 2024-08-07
> **Rebuttal by Authors Part 2**
>
> > **Q1**. The effect of removing stop-words
>
> In Table 7, removing stop-words leads to a decrease in classification accuracy, for example, the linear probing accuracy decreases from 76.0 to 75.7 (-0.3), and the zero-shot accuracy reduces from 61.7 to 61.0 (-0.7). Therefore, we conclude that stopwords could help the vision encoder. This might be explained by the fact that stop-words also carry some useful visual information. For example, 'she', 'her', and 'him' can indicate a person's gender; 'on', 'off', 'above', 'below', 'up', and 'down' can indicate operational status or position; '@' is likely to indicate an email address.
>
> Stop words and punctuation are listed below.
> `['i', 'me', 'my', 'myself', 'we', 'our', 'ours', 'ourselves', 'you', "you're", "you've", "you'll", "you'd", 'your', 'yours', 'yourself', 'yourselves', 'he', 'him', 'his', 'himself', 'she', "she's", 'her', 'hers', 'herself', 'it', "it's", 'its', 'itself', 'they', 'them', 'their', 'theirs', 'themselves', 'what', 'which', 'who', 'whom', 'this', 'that', "that'll", 'these', 'those', 'am', 'is', 'are', 'was', 'were', 'be', 'been', 'being', 'have', 'has', 'had', 'having', 'do', 'does', 'did', 'doing', 'a', 'an', 'the', 'and', 'but', 'if', 'or', 'because', 'as', 'until', 'while', 'of', 'at', 'by', 'for', 'with', 'about', 'against', 'between', 'into', 'through', 'during', 'before', 'after', 'above', 'below', 'to', 'from', 'up', 'down', 'in', 'out', 'on', 'off', 'over', 'under', 'again', 'further', 'then', 'once', 'here', 'there', 'when', 'where', 'why', 'how', 'all', 'any', 'both', 'each', 'few', 'more', 'most', 'other', 'some', 'such', 'no', 'nor', 'not', 'only', 'own', 'same', 'so', 'than', 'too', 'very', 's', 't', 'can', 'will', 'just', 'don', "don't", 'should', "should've", 'now', 'd', 'll', 'm', 'o', 're', 've', 'y', 'ain', 'aren', "aren't", 'couldn', "couldn't", 'didn', "didn't", 'doesn', "doesn't", 'hadn', "hadn't", 'hasn', "hasn't", 'haven', "haven't", 'isn', "isn't", 'ma', 'mightn', "mightn't", 'mustn', "mustn't", 'needn', "needn't", 'shan', "shan't", 'shouldn', "shouldn't", 'wasn', "wasn't", 'weren', "weren't", 'won', "won't", 'wouldn', "wouldn't"]`
>
> `{'{', '+', '(', '$', '}', '!', '%', '\\', '<', ';', '|', ']', '"', "'", ',', '&', '=', ')', '_', '^', '~', '#', '@', '.', '[', '*', '?', ':', '/', '>', '-'}`
>
> > **Q2**. Online IDF
>
> **A2**. To implement online IDF, we set up two global variables to track the number of seen samples (N) and the occurrence count of each subword. As training proceeds, these values will gradually approach the true word frequency of the entire dataset. The advantage of online IDF is its ease of use, allowing the code to be directly transferred to new datasets. We also conducted experiments to compare recognition accuracy using online IDF and offline IDF. The zero-shot accuracy with online IDF drops by about 0.3, and the linear probing accuracy drops by about 0.1.
>
> > **Q3**. The analysis of classification results
>
> **A3**. We evaluated the performance of CLIP and Superclass on the 1000 classes in ImageNet. Among them, Superclass performs better in 502 classes, while CLIP outperforms in 270 classes, and they perform equally well in the remaining classes. We also extracted the top 10 classes with the largest performance differences and displayed them in the table below. We do not find any obvious patterns, but it is possible that Superclass and CLIP can complement each other. This is an interesting direction for future research.
>
> | Class name | muzzle | music speaker | yellow garden spider | cardboard box / carton | Carolina anole | black-footed ferret | monitor | missile | cricket insect | garter snake |
> |------------|--------|---------------|----------------------|------------------------|----------------|---------------------|---------|---------|----------------|--------------|
> | CLIP       | 0.72   | 0.68          | 0.82                 | 0.66                   | 0.66           | 0.62                | 0.36    | 0.44    | 0.58           | 0.82         |
> | SuperClass | 0.88   | 0.84          | 0.96                 | 0.80                   | 0.81           | 0.76                | 0.50    | 0.58    | 0.70           | 0.94         |
>
> | Class name | European polecat | oxygen mask | parallel bars | ox   | promontory | English Setter | stethoscope | split-rail fence | rotisserie | cassette player |
> |------------|------------------|-------------|---------------|------|------------|----------------|-------------|------------------|------------|-----------------|
> | CLIP       | 0.64             | 0.78        | 0.82          | 0.68 | 0.62       | 0.92           | 0.94        | 0.88             | 0.98       | 0.48            |
> | SuperClass | 0.44             | 0.64        | 0.70          | 0.58 | 0.52       | 0.82           | 0.84        | 0.78             | 0.88       | 0.38            |

---

> ### Comment · Reviewer_bBvT · 2024-08-08
>
> At its core, this paper introduces a new way to pre-train open-world Vision-Language foundational models, it's simple and elegant and seems to have maintained the zero-shot capabilities that CLIP offers.
>
> But I believe that using such a model might not be suited for generation, where one may want a text encoder to infer the relationships between words, i.e. "A dog over a cat" vs "A cat over a dog".
>
> Additionally, I believe that such a technique can also be used to clean the pertaining data of VLMs, a problem that LAION has faced in the past [1].
>
> [1] https://cyber.fsi.stanford.edu/news/investigation-finds-ai-image-generation-models-trained-child-abuse
>
> Thanks for the comprehensive answers to my questions, I will increase the score for soundness in my review.

---

> > ### Author Response · Authors · 2024-08-13
> >
> > Thank you for your insightful feedback and for recognizing the strengths of our paper. We appreciate your thoughtful comments on the potential applications and limitations of our model. Your suggestions provide valuable directions for future work.
> >
> > Thank you again for your comprehensive review and for increasing the score. Your support and recognition are greatly appreciated.

---

### Official Review · Reviewer_3a7k · 2024-07-04

**Soundness:** 2
**Presentation:** 3
**Contribution:** 3
**Rating:** 6
**Confidence:** 3

**Summary:**

This paper proposes a simple classification-based vision-language pretraining method. The proposed SuperClass approach directly uses an off-the-shelf subword-level tokenizer to obtain the classification labels from raw text, without requiring any preprocessing. Then, the vision encoder is trained by optimizing the multi-label softmax loss, with Inverse Document Frequency (IDF) as the weight of each label. SuperClass achieves promising performance on classification and various vision-language tasks. Ablation experiments are conducted to validate the impact of different design choices.

**Strengths:**

1. The proposed method in this paper reveals the potential of classification in vision-language pretraining, which provides empirical evidence for further researching.
2. With a simple framework that requires no data preprocessing, SuperClass enables large-scale training on paired image-text data. Compared to previous classification-based pretraining methods, the proposed approach demonstrates greater practical applicability.
3. SuperClass is training-efficient by removing the need for a text encoder, and the extensive experimental results demonstrate the effectiveness and scalability of the proposed approach.

**Weaknesses:**

1. The robustness of the proposed method to different model types remains unclear. All the experiments in this paper use ViT as the vision encoder, and there is no evidence to demonstrate the effectiveness of the SuperClass on other encoder architectures, such as ResNet.
2. The ablation experiment on different classification losses is only conducted on classification tasks, using a ViT-B/16 backbone and 512M seen data samples. It is important to demonstrate the robustness of the softmax loss on a broader range of vision-language tasks, beyond just classification. Furthermore, the impact of the choice of loss function on the scalability of the proposed method is not discussed.

**Questions:**

Please refer to the weaknesses.

**Limitations:**

The authors discuss some of the limitations of their work in Section 5. But I would like them to consider some of my concerns above.

---

> ### Author Rebuttal · Authors · 2024-08-07
>
> > **W1**: The robustness of the proposed method to different model types remains unclear.
>
> **A1.** Thank you for the valuable advice. To evaluate the robustness of our proposed method across different model types, we selected two representative convolution-based networks: ResNet50 and ConvNext-Tiny. We compare SuperClass against CLIP for ImageNet zero-shot (LiT) and linear probing classification, as shown in the table below. All experiments were conducted with a batch size of 16k and 1.28B seen samples.
>
> | Method     | Backbone      | Zero-shot | Linear Probing |
> |------------|---------------|-----------|----------------|
> | CLIP       | RN-50         | 60.73     | 70.28          |
> | Superclass | RN-50         | 62.81     | 71.92          |
> | CLIP       | ConvNext-tiny | 59.94     | 70.35          |
> | Superclass | ConvNext-tiny | 62.85     | 72.33          |
>
> We observe that SuperClass surpasses CLIP in all settings by a clear margin, ranging from 1.64 to 2.91. These results demonstrate that the superiority of SuperClass over CLIP is robust across different model architectures.
>
> > **W2**. The robustness of softmax loss on a broader range of vision-language tasks
>
> **A2**. Thank you for the advice. We further compare the sofrmax loss against different losses on various vision-language tasks. We selected three model sizes (ViT-S/16, ViT-B/16 and ViT-L/16) and trained them on Datacomp-1b with 512 million seen samples. The detailed results are presented in the following table. We observe that softmax is the best-performing loss function across different vision-language tasks and model sizes.
>
> |   Loss  |   dataset   | Seen Sample | Backbone | IN 0-shot | Linear Prob | VQAv2 (val) |  GQA  | VizWiz (val) | TextVQA (val) | SciQA (Img) | MMBench (en/cn) |    MME (P/C)   |  POPE |  MMMU | SEEDBench |
> |:-------:|:-----------:|:-----------:|:--------:|:---------:|:-----------:|:-----------:|:-----:|:------------:|:-------------:|:-----------:|:---------------:|:--------------:|:-----:|:-----:|:---------:|
> | Twoway  | Datacomp-1b | 512M        | vit-S/16 | 50.6      | 66.3        |    65.44    | 55.65 |     43.5     |     15.13     |    66.44    |   51.71/40.72   | 1278.42/335.35 | 79.81 | 34.41 |   50.04   |
> | BCE     | Datacomp-1b | 512M        | vit-S/16 | 47.7      | 65.8        |    64.49    | 55.63 |     48.15    |     13.27     |    65.99    |   50.08/39.86   | 1282.30/323.92 | 79.87 |  35.3 |   50.12   |
> | ASL     | Datacomp-1b | 512M        | vit-S/16 | 48.3      | 66          |    64.58    | 55.49 |     44.96    |     13.44     |    65.59    |   50.77/40.80   | 1290.92/315.15 | 80.46 |  35.1 |   49.98   |
> | Softmax     | Datacomp-1b | 512M        | vit-S/16 | 51.7      | 67          |     65.6    | 56.03 |     43.29    |     16.61     |    64.65    |   49.91/41.92   | 1315.08/306.78 | 81.46 |  35.8 |   51.03   |
> | Twoway  | Datacomp-1b | 512M        | vit-B/16 | 59.7      | 74.8        |    68.05    | 57.79 |     47.35    |     22.09     |    66.63    |   54.55/46.04   | 1350.92/335.71 | 82.51 |  36.8 |   53.02   |
> |  Margin | Datacomp-1b | 512M        | vit-B/16 | 58.1      | 73.5        |    67.08    | 56.67 |     44.17    |     17.87     |     64.9    |   53.43/42.95   | 1341.75/312.14 | 81.71 |  34.7 |   52.32   |
> | BCE     | Datacomp-1b | 512M        | vit-B/16 | 58.5      | 73.6        |    67.35    | 56.91 |     49.2     |     18.34     |    64.95    |   52.49/43.47   | 1327.92/332.14 | 81.89 |  36.7 |   52.76   |
> | ASL     | Datacomp-1b | 512M        | vit-B/16 | 58.7      | 73.8        |    67.59    | 57.02 |     47.02    |     19.01     |    65.44    |   54.72/46.39   |  1345.54/357.5 | 81.93 |  35.3 |   52.75   |
> | Softmax     | Datacomp-1b | 512M        | vit-B/16 | 60.8      | 75.6        |    68.08    | 57.27 |     47.6     |     23.73     |    65.44    |   54.55/46.13   | 1310.65/335.00 | 82.58 |  34.6 |   52.53   |
> | Twoway  | Datacomp-1b | 512M        | vit-L/16 | 66.7      | 78.3        |     70.2    | 58.36 |     46.25    |     27.21     |    64.35    |   57.30/48.62   | 1365.98/315.00 | 82.97 |   36  |   53.87   |
> | BCE     | Datacomp-1b | 512M        | vit-L/16 | 64.9      | 77.2        |    69.56    | 57.93 |     48.62    |      24.9     |     64.3    |   57.64/47.33   | 1316.55/355.71 | 83.17 |  35.1 |   54.01   |
> | ASL     | Datacomp-1b | 512M        | vit-L/16 | 66.1      | 77.6        |    69.71    | 58.43 |     51.16    |     25.49     |    65.49    |   58.41/49.48   | 1389.29/330.35 | 83.51 |  34.6 |   54.07   |
> | Softmax     | Datacomp-1b | 512M        | vit-L/16 | 68.3      | 80.1        |    70.27    | 58.03 |     48.98    |     28.87     |    67.03    |   57.30/49.14   | 1334.33/366.42 | 83.36 |  35.4 |   54.41   |
>
> > **W3**. The scaling properties of loss function
>
> **A3**. Due to the limited time for rebuttal, we only ablated the effect of different losses with different model sizes. As the model size increases, softmax consistently achieves the best accuracy while also demonstrating equal or better scalability compared to other losses.
>
> |         | ViT-S/16 | ViT-S/16 | ViT-B/16 | ViT-B/16 | ViT-L/16 | ViT-L/16 |
> |:-------:|:--------:|:--------:|:--------:|:--------:|:--------:|:--------:|
> |         |    ZS    |    LP    |    ZS    |    LP    |    ZS    |    LP    |
> |  Twoway |   50.6   |   66.3   |   59.7   |   74.8   |   66.7   |   78.3   |
> |  Margin |   47.4   |   65.6   |   58.1   |   73.5   |   64.6   |   76.9   |
> |   BCE   |   47.7   |   65.8   |   58.5   |   73.6   |   64.9   |   77.2   |
> |   ASL   |   48.3   |   66.0   |   58.7   |   73.8   |   66.1   |   77.6   |
> | Softmax |   51.7   |   67.0   |   60.8   |   75.6   |   68.3   |   80.1   |

---

> > ### Comment · Reviewer_3a7k · 2024-08-08
> >
> > Thank you for the response. I will maintain my score at this stage

---

> > > ### Author Response · Authors · 2024-08-13
> > >
> > > Thank you for your quick response and for taking the time to review our paper. We appreciate your feedback and are grateful for your recognition of our work.

---

### Official Review · Reviewer_W9bX · 2024-07-06

**Soundness:** 2
**Presentation:** 3
**Contribution:** 3
**Rating:** 6
**Confidence:** 5

**Summary:**

This paper explores a new direction to pretrain vision backbones using large scale image-text pairs for learning visual representations which are suitable to various downstream tasks. More specifically, this work proposes a classification based objective function as an effective alternative to CLIP's standard cross-modality similarity based constrastive loss. The proposed model SuperClass uses a image encoder to map image to token probabilities having a head size equal to CLIP's tokenizer vocabulary. The texts are converted into labels using weighted tokenizer. Both CLIP baseline and SuperClass are pretrained on datacomp image-text pairs dataset and evaluation results across various vision and vision-language benchmarks are reported. The proposed method shows greater efficiency due to being text-encoder free and also performs favorably well over previous approaches. Extensive ablation studies are performed to justify the design choices made in the paper.

**Strengths:**

**Strengths:**
1) The idea of pretraining large scale vision models using classification objectives is very motivating, as it provides advantages over image-text contrastive loss such as compute efficiency, disentangling the role of text embeddings etc.
2) The proposed method is simple and effective.
3) The experimental results are favorable for SuperClass against its direct baseline CLIP.
4) The choice of the components in the proposed method such as IDF based weighting, loss function and tokenization has been validated in the paper via ablation studies.
5) Paper is easy to read and understand.

**Weaknesses:**

**Weaknesses**
1) In my understanding, one of the weaknesses of this work is lack of comparisons with related works. For example, the only method with which SuperClass is compared with is CLIP which is only a baseline. I believe there should be comparisons with other related SOTA works.
2) The proposed approach might not be capable of doing multi-label classification or zero-shot classification as compared to other competitors such as RAM, CLIP etc. Also CLIP has out of the box additional features such as prompt ensembling, zero-shot segmentation [1] etc which might be not present in SuperClass. This would question how it can comprehensively show effectiveness over CLIP and other vision-language models.
3) It is not clear how the proposed method learns suitable representations when using the subwords as labels which does not correspond to any visual concept. I believe there is a bit of analysis missing in the manuscript.
4) The results for the zeroshot and linear probing in Tab1 and Tab 2 of the main paper are different for the same model. It is unclear if why the results are different for same model.


[1] Extract Free Dense Labels from CLIP (ECCV 2022)

**Questions:**

Please refer to the weaknesses section for the questions. I will highly recommend the authors submit a rebuttal response. I will be happy to reconsider my final scores based on the response from the authors.

**Limitations:**

Yes the authors have adequately addressed the limitations in the paper.

---

> ### Author Rebuttal · Authors · 2024-08-07
>
> > **W1**: Comparisons with other related SOTA works
>
> **A1**. Thanks for the advice. We have added more SOTA methods for comparison, including self-supervised learning methods (MoCov3, Dinov1&v2, MAE, BEiT, CAE) and weakly-supervised methods (CatLIP, Cappa). Additionally, we have evaluated more downstream tasks and datasets, such as semantic segmentation on ADE20K and instance segmentation on COCO. Please refer to our response to Reviewer1(aefc) and Reviewer4(bBvT) for more results.
>
>
> **Linear probing on ImageNet-1K**
>
> | Method | Pre-training Data | ViT-Base | ViT-Base  | ViT-Large | ViT-Large |
> |:---:|:---:|:---:|:---:|:---:|:---:|
> |  |  | #Seen Samples | Top-1 (%) | #Seen Samples | Top-1 (%) |
> | contrastive or clustering based |  |  |  |  |  |
> | MoCov3 |  IN1K | 400M | 76.7 | 400M | 77.6 |
> | DINO | IN1K | 512M | 78.2 | - | - |
> | iBoT | IN22K | 400M | 79.5 | 256M | 81.0 |
> | DINOv2 | LVD-142M | 1.28B | - | 1.92B | 84.5 |
> | reconstruction based |  |  |  |  |  |
> | BEiT | D250M+IN22K | 1B | 56.7 | 1B | 73.5 |
> | SimMIM | IN1K | 1B | 56.7 | - | - |
> | CAE | D250M | 2B | 70.4 | 2B | 78.1 |
> | MAE | IN1K | 2B | 68.0 | 2B | 75.8 |
> | language-image pretraining based |  |  |  |  |  |
> | CLIP | WIT400M | 12.8B | 78.5 | 12.8B | 82.7 |
> | Cappa | WebLI-1B | - | - | 9B | 83.0 |
> | OpenCLIP | Datacomp-1B | - | - | 12.8B | 83.9 |
> | Superclass | Datacomp-1B | 12.8B | 80.2 | 12.8B | 85.0 |
> | Superclass | Datacomp-1B | 1.28B | 78.7 | 1.28B | 82.6 |
> | Superclass | Datacomp-1B | 512M | 75.6 | 512M | 80.5 |
>
> **Instance segmentation and semantic segmentation**
>
> Results of instance segmentation are obtained by using Mask R-CNN on COCO with an input resolution of 1024×1024. Semantic segmentation results are obtained by using UperNet on ADE20K with an input resolution of 512×512.
> | Method | #Seen Samples | Semantic Segmentation mIoU | Instance Segmentation APmask |
> |:---:|:---:|:---:|:---:|
> | Supervised | - | 49.9 | 43.9 |
> | MoCov3 | 400M | 49.1 | 44.0 |
> | BEiT | 400M | 53.3 | 47.1 |
> | CAE | 2B | 54.7 | 47.6 |
> | MAE | 2B | 53.6 | 47.2 |
> | CLIP | 12.8B | 57.9 | 48.3 |
> | Superclass | 1.28B | 56.2 | 48.1 |
> | Superclass | 12.8B | 58.4 | 49.0 |
>
>
> **VLM downstream tasks**
> | Model | Size | VQAv2 | GQA | VizWiz | TextVQA | SciQA | MME | MMBench | PoPE | MMMU |
> |---|:---:|:---:|:---:|:---:|:---:|:---:|:---:|:---:|:---:|:---:|
> | openCLIP | ViT-Large | 74.54 | 61.03 | 50.47 | 38.16 | 67.33 | 1434.86/268.92 | 60.73 | 85.52 | 35.9 |
> | MAE | ViT-Large | 63.5 | 54.58 | 50.22 | 11.55 | 64.75 | 1175.04/343.92 | 42.44 | 80.69 | 35.7 |
> | DINOv2 | ViT-Large | 73.32 | 61.87 | 49.15 | 14.08 | 64.9 | 1335.61/296.78 | 57.9 | 86.24 | 35.3 |
> | Superclass | ViT-Large | 75.24 | 60.96 | 54.33 | 39.2 | 66.09 | 1371.32/321.78 | 63.14 | 85.69 | 36 |
> openCLIP and Superclass are trained with 12.8B seen samples.
>
>
> > **W2**: The proposed approach might not be capable of doing multi-label classification or zero-shot classification ...how it can comprehensively show effectiveness over CLIP and other vision-language models.
>
> **A2**. Our paper focuses on the issue of vision encoder pretraining, specifically whether the pre-trained model can perform better in vision-language models. Therefore, the native zero-shot capability may not be the primary property we consider.
> However, we can use Lock Image Tuning(LiT)[69] to equip our pre-trained vision backbone with zero-shot classification and retrieval capabilities. After LiT, we also could do zero-shot segmentation using MaskCLIP[a]. The experimental results show that SuperClass could achieve much better performance on the Pascal context and COCO stuff dataset.
> | Method     | Backbone | #Seen sample | PASCAL Context | COCO Stuff |
> |------------|----------|--------------|----------------|------------|
> | CLIP       | ViT-B/16 | 1.28B        | 16.2           | 8.7        |
> | Superclass | ViT-B/16 | 1.28B        | 20.2(+4.0)     | 13.2(+4.5) |
>
> [a] Extract Free Dense Labels from CLIP (ECCV 2022)
>
>
>
> > **W3**:  using the subwords as labels does not correspond to any visual concept
>
> **A3**. Firstly, it is important to note that a single object class can be mapped to one or multiple subwords. Here is a detailed explanation of how our method works in both scenarios:
> 1. Single Subword Mapping:
>   - When a class is mapped to a single subword, the process is similar to traditional classification tasks. The model learns to associate the subword with the corresponding visual patterns.
>   - This is akin to standard classification where each class is represented by a unique label, and the model learns the association between the label (subword) and the visual features.
> 2. Multiple Subword Mapping:
>   - When a class is mapped to multiple subwords, our optimization objective is to maximize the co-occurrence probability of subwords that belong to the same class.
>   - This means the model learns to associate multiple subwords with the corresponding visual patterns, effectively capturing the relationship between subwords and the visual concept they represent.
>
>
>
> > **W4**: The results for the zero-shot and linear probing in Tab1 and Tab 2 of the main paper are different for the same model.
>
> **A4**. Because the models are trained with different #seen samples. In Table 1 and Table 2, we use the ViT-L/16 as the backbone. The models in Table 1 are trained with 12.8B seen samples. In Table 2, we study the effect of different seen samples. The models are trained with 128M, 512M, and 1.28B seen samples, respectively.

---

> > ### Comment · Reviewer_W9bX · 2024-08-09
> > **Thank you for providing the rebuttal response**
> >
> > Dear Authors,
> >
> > Thank you for providing the rebuttal response.
> >
> > While the proposed SuperClass method is not natively zero-shot, it shows other various flexibilities as demonstrated in the rebuttal. More importantly, this is a new pretraining style for vision backbones rather than a variant of CLIP. Honestly, the current pitch of the paper puts too much emphasis on CLIP (instead of advocating for vision backbone pretraining), and that is why many of the concerns from the reviewers are about comparisons with CLIP-like models.
> >
> > In the end, I believe this paper would allow the research community to improve vision backbone pretraining and have good insights. Therefore I will increase my score to weak accept, and hope that the paper is accepted.
> >
> > For the final version, I strongly recommend the authors to include all the rebuttal discussions in the main paper, and also put more emphasis on vision backbone pretraining, so that no ambiguities are left.

---

> > > ### Author Response · Authors · 2024-08-13
> > >
> > > Thank you for your valuable reviews and feedback.
> > >
> > > We appreciate your insights on emphasizing the new pretraining style for vision backbones rather than comparing it predominantly with CLIP. We will ensure that the final version of the paper includes all the rebuttal discussions and places greater emphasis on vision backbone pretraining to eliminate any ambiguities.
> > >
> > > Thank you again for your constructive comments and for increasing your score to a weak accept. We hope that our paper will contribute positively to the research community.

---

### Official Review · Reviewer_aefc · 2024-07-08

**Soundness:** 2
**Presentation:** 3
**Contribution:** 2
**Rating:** 3
**Confidence:** 4

**Summary:**

This paper introduces a multi-label classification pre-training style for visual image encoder pre-training.

**Strengths:**

The proposed method is straightforward.

**Weaknesses:**

- The zero-shot capacity of such a multi-label pre-trained model is not well demonstrated.
- The paper lacks a comprehensive comparision with weakly-supervised or unsupervised visual encoder methods. Only comparing with CLIP is not enough.
- Some arguments about previous so-called "bag-of-word classification" pre-trained methods may not be correct.

**Questions:**

- 1. The greatest advantage of CLIP is its zero-shot capacity across various downstream tasks. The zero-shot ability of SuperClass is not well-demonstrated on downstream tasks like zero-shot text-image retrieval, zero-shot text-video retrieval, and zero-shot STR. The reviewer thinks the authors should include an analysis of the zero-shot ability like the original CLIP paper.

- 2. The baselines should not be CLIP-style pre-trained vision language models. The proposed method aims to pre-train a vision transformer. The baseline should be other weakly-supervised methods or self-supervised methods like MAE, etc. The authors should compare these methods in terms of training efficiency and performance of transfer learning. These methods also show good transfer learning ability, for example, MAE has good performance of transfer learning on COCO, while it is only pre-trained on ImageNet-1K.

- 3. Some arguments about previous so-called "bag-of-word classification" pre-trained methods may not be correct. In the introduction, the authors claim " However, these methods fail to gain popularity from the community, as most of the experiments are conducted on a small scale and there is no evidence showing their scalability to data size and model size in comparison to CLIP ". Nevertheless, CatCLIP[1] already conducts experiments with CLIP-H on DataComp-1.3B.
    - 3.1. The authors should also compare with these "bag-of-word classification" pre-trained methods.

- 4. The authors say "All experiments were carried out on an A100 GPU equipped with 80GB of memory." Could the authors provide training times for the experiments in Table 1? Besides, are experiments in Table 2 also trained on an A100 80G? If so, could the authors provide the training time?

[1] https://arxiv.org/pdf/2404.15653

**Limitations:**

Yes.

---

> ### Author Rebuttal · Authors · 2024-08-07
>
> >**Q1** Zero-shot capacity.
>
> **A1.** We thank the reviewer for triggering the discussion on zero-shot abilities of CLIP and our model. Honestly, our model does not come with trivial zero-shot image-text retrieval usage. However, we can enable this behavior by levering LiT [a], which learns a text encoder by keeping the image encoder fixed. As shown by the following table, we test the zero-shot ability of our model on 10 datasets. Superclass beats openCLIP on seven of the datasets. It is worth noting that SuperClass uses a ViT large model with patch size 16, while the baseline method adopts a patch size of 14. This makes the superiority of SuperClass over openCLIP even clearer.
>
> | Case       | Model    | Data        | Seen samples | IN-1K | imagenet-v2 | imagenet-r | imagenet-a | imagenet-sketch | GTSRB | Rendered SST2 | ObjectNet | SUN397 | Country211 |
> |------------|----------|-------------|--------------|-------|-------------|------------|------------|-----------------|-------|---------------|-----------|--------|------------|
> | openCLIP   | ViT-L/14 | Datacomp-1B | 12.8B        | 79.2  | 72.1        | 90.8       | 69.6       | 68.0            | 58.5  | 61.0          | 74.3      | 74.3   | 31.6       |
> | SuperClass | ViT-L/16 | Datacomp-1B | 12.8B        | 79.7  | 72.4        | 91.6       | 68.8       | 70.6            | 58.5  | 61.6          | 73.9      | 73.8   | 32.3       |
>
> Moreover, we'd like to emphasize that there is another increasingly more important application of the CLIP model. Specifically, CLIP is predominantly the default vision encoder in existing vision-language models (e.g., LLaVA, BLIP). We show that when combined with a large language model, SuperClass is able to substantially improve performance over CLIP. Please refer to Table 2 and Table 3 on the paper for details.
>
> [a] Zhai, Xiaohua, et al. "Lit: Zero-shot transfer with locked-image text tuning." CVPR. 2022.
>
> >**Q2** The baseline should be other weakly-supervised methods or self-supervised methods like MAE, etc
>
> **A2.** Thanks for the suggestion. We have added more SOTA methods for comparison, including self-supervised methods (MoCov3, Dinov1&v2, MAE, BEiT, CAE) and weakly-supervised methods (CLIP, Cappa). Additionally, we have evaluated more downstream tasks and datasets, such as semantic segmentation on ADE20K and instance segmentation on COCO.
> Finally,  Following the LLaVA setup, we combine frozen CLIP models, self-supervised models, and SuperClass
> models with the pre-trained Vicuna-V1.5-7B and perform downstream tasks.
> The experimental results demonstrate that the proposed method could achieve better than self-supervised ViT pre-training methods, like Dinov2, and weakly-supervised methods, like CLIP.
>
> **Linear probing on ImageNet-1K**
>
> | Method | Pre-training Data | ViT-Base | ViT-Base  | ViT-Large | ViT-Large |
> |:---:|:---:|:---:|:---:|:---:|:---:|
> |  |  | #Seen Samples | Top-1 (%) | #Seen Samples | Top-1 (%) |
> | contrastive or clustering based |  |  |  |  |  |
> | MoCov3 |  IN1K | 400M | 76.7 | 400M | 77.6 |
> | DINO | IN1K | 512M | 78.2 | - | - |
> | iBoT | IN22K | 400M | 79.5 | 256M | 81.0 |
> | DINOv2 | LVD-142M | 1.28B | - | 1.92B | 84.5 |
> | reconstruction based |  |  |  |  |  |
> | BEiT | D250M+IN22K | 1B | 56.7 | 1B | 73.5 |
> | SimMIM | IN1K | 1B | 56.7 | - | - |
> | CAE | D250M | 2B | 70.4 | 2B | 78.1 |
> | MAE | IN1K | 2B | 68.0 | 2B | 75.8 |
> | language-image pretraining based |  |  |  |  |  |
> | CLIP | WIT400M | 12.8B | 78.5 | 12.8B | 82.7 |
> | Cappa | WebLI-1B | - | - | 9B | 83.0 |
> | OpenCLIP | Datacomp-1B | - | - | 12.8B | 83.9 |
> | Superclass | Datacomp-1B | 12.8B | 80.2 | 12.8B | 85.0 |
> | Superclass | Datacomp-1B | 1.28B | 78.7 | 1.28B | 82.6 |
> | Superclass | Datacomp-1B | 512M | 75.6 | 512M | 80.5 |
>
> **Instance segmentation and semantic segmentation**
>
> Results of instance segmentation are obtained by using Mask R-CNN on COCO with an input resolution of 1024×1024. Semantic segmentation results are obtained by using UperNet on ADE20K with an input resolution of 512×512.
> | Method | #Seen Samples | Semantic Segmentation mIoU | Instance Segmentation APmask |
> |:---:|:---:|:---:|:---:|
> | Supervised | - | 49.9 | 43.9 |
> | MoCov3 | 400M | 49.1 | 44.0 |
> | BEiT | 400M | 53.3 | 47.1 |
> | CAE | 2B | 54.7 | 47.6 |
> | MAE | 2B | 53.6 | 47.2 |
> | CLIP | 12.8B | 57.9 | 48.3 |
> | Superclass | 1.28B | 56.2 | 48.1 |
> | Superclass | 12.8B | 58.4 | 49.0 |
>
>
> **VLM downstream tasks**
> | Model | Size | VQAv2 | GQA | VizWiz | TextVQA | SciQA | MME | MMBench | PoPE | MMMU |
> |---|:---:|:---:|:---:|:---:|:---:|:---:|:---:|:---:|:---:|:---:|
> | openCLIP | ViT-Large | 74.54 | 61.03 | 50.47 | 38.16 | 67.33 | 1434.86/268.92 | 60.73 | 85.52 | 35.9 |
> | MAE | ViT-Large | 63.5 | 54.58 | 50.22 | 11.55 | 64.75 | 1175.04/343.92 | 42.44 | 80.69 | 35.7 |
> | DINOv2 | ViT-Large | 73.32 | 61.87 | 49.15 | 14.08 | 64.9 | 1335.61/296.78 | 57.9 | 86.24 | 35.3 |
> | Superclass | ViT-Large | 75.24 | 60.96 | 54.33 | 39.2 | 66.09 | 1371.32/321.78 | 63.14 | 85.69 | 36 |
> openCLIP and Superclass are trained with 12.8B seen samples.
>
> >**Q3** Some arguments about previous so-called "bag-of-word classification"
>
> **A3.** Thanks for pointing it out. We will rewrite the description here. CatLIP [46] is a concurrent work. We have already cited and compared it in the paper.

---

> > ### Comment · Reviewer_aefc · 2024-08-12
> > **Discussion**
> >
> > Dear authors:
> >
> > Thanks for your time, efforts, and response.
> >
> > - **Q1** So far, we only see zero-shot performance on classification tasks. It seems that CLIP training style shows broader zero-shot ability in different downstream tasks.
> >
> > - **Q2** From the comparison, I do not see superioty of the proposed pre-trained method compared to self-supervised learning especially the proposed method is actually weakly-supervised. Table 4 in DINOv2 provide a more comprehensive comparison.
> > It seems DINOv2 also have better transfer learning results in its Table 10.
> >
> > Overall, I do not think the proposed methods show enough superiorty compared to CLIP and existing weakly-supervised/self-supervised methods, especially when there are some similar works like CatLIP.
> >
> > At this time, I still hold my original score.
> >
> > [1] https://arxiv.org/pdf/2304.07193

---

> ### Author Response · Authors · 2024-08-07
> **Rebuttal by Authors Part 2**
>
> >**Q4** Compared with these "bag-of-word classification" pre-trained methods
>
> **A4.** We have included the comparison with other classification-based methods, like CatLIP in the subsection Word-level tokenizer vs. Subword-level tokenizer. The word-level tokenizer is used in CatLIP [46], which carefully selected approximately 40,000 "gold labels" from the datacomp-1B dataset. Aside from the tokenizer being different, all models are trained under the same settings. The results of Table 4 show that with the increasing size of the model, the subword-level tokenizer gradually outperforms the word-level tokenizer, whether in classification tasks or vision & language tasks.
> We also provide the results of finetuning on ImageNet-1k in the below Table. The superclass could achieve better performance than CatLIP.
> | Model | Pretraining |  ImageNet-1k Fine-tuning |
> |---|---|---|
> | OpenCLIP ViT-L/14 | Datacomp-1B | 87.4 |
> | CatLIP ViT-L/16* | Datacomp-1B | 86.5 |
> | Superclass ViT-L/16 | Datacomp-1B | 87.8 |
> *number from the paper
>
> >**Q5** GPU usage
>
> **A5.** This is a typo. What we meant to say is all experiments were carried out on 80G A100 GPUs. We will fix it in the revised version.

---

> ### Author Response · Authors · 2024-08-13
> **Rebuttal by Authors Part 3**
>
> Thanks for your reviews and feedback.
> >**Q1:** So far, we only see zero-shot performance on classification tasks. It seems that CLIP training style shows broader zero-shot ability in different downstream tasks.
>
> **A1:**  We have evaluated zero-shot retrieval on COCO dataset and zero-shot segmentation on Pascal context and COCO stuff (see more details in the response to Reviewer W9bX). The experimental results show that Superclass could achieve competitive performance compared to CLIP.
> | Case | Model | COCO Image-to-Text |  | COCO Text-to-Image |  |
> |:---:|:---:|:---:|:---:|:---:|:---:|
> |  |  | R@1 | R@5 | R@1 | R@5 |
> | openCLIP | ViT-Large | 62.6 | 84.2 | 46.9 | 70.7 |
> | SuperClass | ViT-Large | 62.1 | 83.3 | 47.1 | 70.7 |
>
> | Method | Backbone | #Seen sample | PASCAL Context | COCO Stuff |
> |---|---|---|---|---|
> | CLIP | ViT-B/16 | 1.28B | 16.2 | 8.7 |
> | Superclass | ViT-B/16 | 1.28B | 20.2(+4.0) | 13.2(+4.5) |
>
> Besides, we also present results of linear probing and fine-tuning on ImageNet-1K. Compared to CLIP, SuperClass outperforms openCLIP on seven out of ten zero-shot classification datasets. Our method achieves a 1.1% higher accuracy in IN-1K linear probing (85.0 vs 83.9). To further demonstrate the transfer ability of our method, we conducted semantic segmentation on ADE20K and instance segmentation on COCO. Moreover, we provide experimental results on various vision & language tasks after integrating with large language models. The results show that our method outperforms CLIP.
>
> We hope that by showcasing our competitive performance in zero-shot retrieval, zero-shot segmentation, linear probing, fine-tuning, semantic segmentation,  instance segmentation, and various vision & language tasks, we can address the reviewer’s concern of "only see zero-shot performance on classification tasks".
>
> >**Q2:** From the comparison, I do not see the superiority of the proposed pre-trained method compared to self-supervised learning, especially since the proposed method is actually weakly-supervised.
>
> **A2:**  Compared to the current SOTA self-supervised model DINOv2, our method achieves a 0.5% higher accuracy in IN-1K linear probing (85.0 vs 84.5). Although SuperClass has seen more samples, our method is very simple and straightforward. DINOv2 adopts a dual-tower structure and adds a bunch of bells and whistles as shown in Table 1.
>
> Furthermore, we would like to remind Reviewer aefc the very important comparison on vision and language tasks. Our method outperforms DINOv2 in 7 out of 9 tasks, and the overall score is significantly higher than that of DINOv2 (Dinov2 54.66 vs. Ours 58.94). Please note that the Dinov2 used here is distilled from the ViT-giant model. These improvements are significant.
>
>
> We thank the reviewer bBvT for the comment, "At its core, this paper introduces a new way to pre-train open-world Vision-Language foundational models," and Reviewer W9bX for noting, "this is a new pretraining style for vision backbones rather than a variant of CLIP," our proposed new pretraining method is distinctly different from previous pretraining approaches. The differences from other most closely related classification-based pretraining methods are also evident: our method does not require manually curated "golden labels" and directly uses tokenized raw text as supervision. In the paper subsection "Word-level tokenizer vs. Subword-level tokenizer," we demonstrate through experiments that our method has better performance and scalability. As Reviewer 3a7k mentioned, "Compared to previous classification-based pretraining methods, the proposed approach demonstrates greater practical applicability."
>
> Overall, we believe we have provided a thorough response to Reviewer aefc's comments and hope that Reviewer aefc will review our feedback in light of the perspectives that align their rating with other reviewers.

---

> ### Comment · Reviewer_aefc · 2024-08-13
> **Discuss**
>
> Dear authors:
>
> Thank you for you response.
>
> - Q1
> ***
> Could you please tell me where the zero-shot 46.9 COCO Text-to-Image result comes from? I check the paper of CLIP[1], the zero-shot image retrieval result is 37.8.
>
> - Q2
> ***
> - CLIP is born in  26 Feb 2021. Many other VLM pre-trained under language supervision emerged, e.g., EVA-CLIP[2]
> - Self-supervised pre-training and weaky-supervised methods are also hot. According to the provided information, the proposed method uses 10x data (12.8B), and it is close to DINOv2(1.92B). If we talk about "performance and scalability", DINOv2 ViT-g/14@448 achieves 86.7% linear probing results on ImageNet-1K.
>
> Overall, the baselines are far from strong. The reviewer does not know why we need such a method when we have many other powerful and versatile pre-trained VLMs/self-supervised large models/weakly supervised large models.
>
> I still retain my score at this time.
>
> Best,
>
> Reviewer aefc
>
> [1] https://arxiv.org/pdf/2103.00020
>
> [2] https://arxiv.org/abs/2303.15389

---

> ### Author Response · Authors · 2024-08-14
> **Rebuttal by Authors Part 4**
>
> We thank reviewer aefc for their timely response.
>
> Regarding Q1, we utilized the open-sourced checkpoint [1] from paper [2], which employs ViT-Large as its backbone and is trained with Datacomp-1B and 12.8 billion seen samples. This serves as a stronger baseline compared to the original OpenAI CLIP.
>
> >**Reviewer aefc commented "The reviewer does not know why we need such a method when we have many other ... pre-trained ... large models" (VLMs/self-supervised/weakly supervised).**
>
> It seems the reviewer suggests that foundational modeling research might be redundant when the absolute performance are weaker compared to a "large model" pretrained in a systematic way. The reviewer appears to prioritize "higher numbers" even when comparing in an apple-to-orange setting. While we acknowledge that a big model with higher numbers is often advantageous from a downstream users' perspective, we believe that a scientific foundational model researcher would in favor of careful ablations rather than chasing higher numbers with biggest models.
>
> For instance, the reviewer argues that SuperClass cannot demonstrate better "performance and scalability" than DINO v2, citing that the DINO v2 *ViT-g 448* model, pretrained on a customized dataset at 448 resolution - superior to our largest *ViT-L 224* model at 224 resolution.
>
> We appreciate the reviewer's observation that our *ViT-L 224* resolution model is less powerful compared to a *ViT-g 448* resolution model. It appears that the reviewer has overlooked various systematic differences - unintentionally or for some other unknown reason - comparing scientific papers in an apple-to-orange manner.
>
> >**The reviewer aefc further suggested "Overall, the baselines are far from strong. "**
>
> The baselines we compared here are CLIP trained with DataComp [2] and DINOv2 [3], which are important milestone papers accepted at NeurIPS 2023 and TMLR 2024, respectively. For some unknown reason, the reviewer believes CLIP, DataComp and DINOv2 are far from strong. We kindly respect the reviewer's false claim.
>
> We would like to wish the reviewer the best, and politely disagree with the reviewer's comments and rating.
>
> We further thank other reviewers for their professional, scientific, objective reviews.
>
> [1] https://huggingface.co/laion/CLIP-ViT-L-14-DataComp.XL-s13B-b90K
>
> [2] DATACOMP: In search of the next generation of multimodal dataset, NeurIPS 2023.
>
> [3] DINOv2: Learning Robust Visual Features without Supervision, TMLR 2024.

---

> ### Comment · Reviewer_aefc · 2024-08-14
> **Discussion**
>
> Thank the authors for the response.
>
> ## 1. "performance and scalability"
> ***
> Well, the authors say that your methods have better "performance and scalability" in https://openreview.net/forum?id=Hd2EOwKItm&noteId=KeV1R5CtDE **A2**.
> So,
> - The reviewer provides examples that previous methods have better "performance and scalability".
> - The proposed method use **10x** data, but is close to DINOv2 with the same parameters.
> -  The reviewer does not even mention similar work CatCLIP[1], which also trains CLIP in a classification way. It also demonstrates better "performance and scalability".
>
> ## 2. It seems the reviewer suggests that foundational modeling research might be redundant when the absolute performance are weaker
> ***
> The authors are trying to mislead readers.
>
> - The paper does not propose something novel, CatCLIP[1] already does the classification pre-training job.
> - The performance/scalability of the paper is no better than recent literature [2,3].
>
> The proposed method does not demonstrate its necessity when there are many other strong self-supervised/weakly-supervised/language supervised (may belong to the weakly-supervised class) methods, so it is reasonable that the reviewer thinks this work may not be necassary for the community.
>
> ## 3. Different data from CLIP paper
>
> The authors do not answer this question. https://openreview.net/forum?id=Hd2EOwKItm&noteId=vqnfjpW7aT Q1.
>
> ```
> Could you please tell me where the zero-shot 46.9 COCO Text-to-Image result comes from? I check the paper of CLIP[1], the zero-shot image retrieval result is 37.8.
> ```
>
>
>
> ## 4.
> Finally, as an independent reviewer, the reviewer thinks he can have his own opinion and rating about the paper.
>
>
> Best,
>
> Reviewer aefc
>
> [1] https://arxiv.org/pdf/2404.15653
>
> [2] https://arxiv.org/abs/2303.15389
>
> [3] DINOv2: Learning Robust Visual Features without Supervision, TMLR 2024.

---

> ### Author Response · Authors · 2024-08-14
>
> Thank the Reviewer aefc for the feedback.
>
> We have addressed all Reviewer aefc's questions in our previous responses, including but not limited to
> - performance and scalability
>
>     > Line #212 subsection "Data Scaling results Line" and #221 subsection "Model scaling results" in the paper
>
>     > The results of Zero-shot classification, VLM downstream tasks ... in https://openreview.net/forum?id=Hd2EOwKItm&noteId=JxjyAuZo5M (Rebuttal by Authors)
>
>     > Few-shot classification in A1 https://openreview.net/forum?id=Hd2EOwKItm&noteId=1hHvBn6szv (Response to Reviewer bBvT)
>
> - comparisons and differences with CatLIP[46]
>
>     > A3 in https://openreview.net/forum?id=Hd2EOwKItm&noteId=JxjyAuZo5M (Rebuttal by Authors) "CatLIP [46] is a concurrent work. We have already cited and compared it in the paper. "
>
>     > Line #69 and Line #233 subsection "Word-level tokenizer vs. Subword-level tokenizer" in the paper
>
>     > Fine-tuning results in A4 https://openreview.net/forum?id=Hd2EOwKItm&noteId=OOhKhswjjl (Rebuttal by Authors Part 2)
>
>     > https://openreview.net/forum?id=Hd2EOwKItm&noteId=KeV1R5CtDE (Rebuttal by Authors Part 3) "The differences from other...classification-based pretraining methods are also evident...manually curated "golden labels"..."
>
> - Performance comparisons with CLIP and Dinov2
>
>     > The results of Zero-shot classification, VLM downstream tasks ... in https://openreview.net/forum?id=Hd2EOwKItm&noteId=JxjyAuZo5M (Rebuttal by Authors)
>
>     > The results of 10-shot classification in A1 of https://openreview.net/forum?id=Hd2EOwKItm&noteId=aVM6gFdSHR (Response to Reviewer bBvT)
>
>     ...
>
> - Different data from CLIP paper
>
>     > https://openreview.net/forum?id=Hd2EOwKItm&noteId=ogRsx4SgwE (Rebuttal by Authors Part 4) "Regarding Q1, we utilized the open-sourced checkpoint [1] from paper [2]...This serves as a stronger baseline compared to the original OpenAI CLIP."
>
> We do not intend to engage in repetitive responses and hope to strengthen and solidify our work based on the reviewers' valuable feedback.
>
> We further thank all the reviewers.

---

### Decision · Program_Chairs · 2024-09-25

**Decision:**

Accept (poster)

**Comment:**

Thanks to the authors for submitting this work. I found the proposed method simple yet elegant, requiring no data preprocessing, and can be applied at large data scales efficiently. The authors conducted comprehensive experimental results, including added results during rebuttal on few-shot/zero-shot tasks and comparison with related self-supervised/ weakly-supervised methods beyond CLIP.

As noted by the reviewers, there are some minor issues in the current papers: (1) similar features to recent work e.g. CatCLIP (however, I noted this is a concurrent work) and (2) the current contribution of vision backbone pretraining (rather than just CLIP-related) is not emphasized enough in the paper.

In conclusion, I recommend accepting this paper and hope the authors will continue to address the above-mentioned points in the revised paper.